# Rehabilitation of Patients with Arthrogenic Muscular Inhibition in Pathologies of Knee Using Virtual Reality

**DOI:** 10.3390/s23229114

**Published:** 2023-11-11

**Authors:** Juan Pablo Flórez Fonnegra, Andrea Carolina Pino Prestan, Lucelly López López, Juan C. Yepes, Vera Z. Pérez

**Affiliations:** 1Facultad de Ingeniería Electrónica, Universidad Pontificia Bolivariana, Medellín 050031, Colombia; juan.florez@upb.edu.co (J.P.F.F.); juancamilo.yepes@upb.edu.co (J.C.Y.); 2Grupo de Investigaciones en Bioingeniería, Universidad Pontificia Bolivariana, Medellín 055031, Colombia; ancapipre@hotmail.com; 3Facultad de Medicina, Universidad Pontificia Bolivariana, Medellín 055031, Colombia; lucelly.lopez@upb.edu.co; 4Grupo de Automática y Diseño A+D, Universidad Pontificia Bolivariana, Medellín 055031, Colombia

**Keywords:** dynamometer, virtual reality, arthrogenic muscle inhibition, electromyography, therapy

## Abstract

Arthrogenic muscle inhibition (AMI) refers to muscular alterations that are generated, producing biomechanical motor control and movement problems, leading to deficiencies in strength and atrophy. Currently, there exist methods that involve virtual reality (VR) and have been well perceived by physiotherapists. The present research measured the potential benefits in terms of therapeutic adherence and speed of recovery, through a comparative analysis in a healthcare provider institution, in Medellín, Colombia, with and without the aid of VR. For this purpose, dynamometry, and surface electromyography (sEMG) signal acquisition tools were used. The treatment involved neuromodulation, ranges of motion and mobility work, strengthening and reintegration into movement, complemented with TENS, NMENS and therapeutic exercise, where the patient was expected to receive a satisfactory and faster adherence and recovery. A group of 15 people with AMI who include at least 15 min of VR per session in their treatment were compared with another group who received only the base treatment, i.e., the control group. Analyzing the variables individually, it is possible to affirm that VR, as a complement, statistically significantly improved the therapeutic adherence in 33.3% for CG and 37.5% for IG. Additionally, it increased strength with both legs, the symmetry between them, and decreased the level of pain and stiffness that is related to mobility.

## 1. Introduction

Currently, injuries involving the skeletal and muscular systems are considered a public health problem, since they represent up to 80% of physiotherapy consultations [1]. Worldwide, a substantial portion of the population experiences these musculoskeletal issues, which can be attributed to several factors, including poor postural habits, occupational activities [2], age, gender [3], participation in sports activities, insufficient musculoskeletal strength, and others. All these factors can result in mobility limitations [4].

Functional assessment techniques have proven their usefulness as complementary medical tests for evaluating the skeletal and muscular systems of patients in various fields: rehabilitation services, traumatology, neurology, occupational medicine, medical services of insurance companies, centers for qualification and assessment of disabilities and orthopedics, among others [5]. In the field of physiotherapy, these techniques are crucial for patients with muscle alterations, biomechanics of motor control, strength deficit and muscle atrophy, all of which can lead to a condition known as athrogenic muscle inhibition (AMI). AMI is a defense mechanism that effects the nervous strength and mobility [6], and refers to muscular alterations that are generated, producing biomechanical motor control and movement problems, leading to deficiencies in strength and atrophy, although the magnitude of the reported activation deficits varies, ranging anywhere from 8% to 45% [7]. This occurs because the damage received is often greater than what said system is capable of tolerating, and before pain occurs in a joint for any reason (trauma, disease), muscle changes occur that lead to biomechanical, motor control and movement compromises.

Although in traditional physiotherapy this biological process is treated with physical mechanisms and intervention from the physiological aspect; it is visualized that sometimes patients do not improve despite many rehabilitation sessions in the established times due to finding physical, psychological, socio-demographic and clinical barriers [8], for which additional techniques are currently implemented that enhance effectiveness and show some additional benefits [9].

These additional techniques may encompass emerging technologies as innovations with the potential to revolutionize the field, influence industries, society and the way people live and work, such as virtual reality, artificial intelligence, or wearable devices. By integrating these emerging technologies, physiotherapists could provide more personalized and innovative solutions, potentially overcoming some of the barriers that traditional approaches may face.

One of the techniques that has made strong inroads in the rehabilitation area is VR, considered as a set of devices that recreate an environment through simulations, allowing an active interaction between the user with a fictitious environment and with important benefits in therapeutic adherence and in the recovery of certain pathologies [10].

The VR technique is gaining more and more strength in the physiotherapy area, a factor that enhances its therapeutic methods by involving VR has been noted in the professional practice [11]. Therefore, this research aims to compare the therapeutic adherence and speed of recovery achieved by patients who undergo VR versus those who do not.

In physiotherapy, the concept of therapeutic adherence is multivariate [12], since it can be associated with the attendance and frequency of sessions, following the advice and prescriptions of the treating professional, the medical-care-provided organization, receptivity to treatment, willingness, and intensity to complete each session, among others [8].

VR has been incorporated into physical therapy practice over the past 20 years as a tool that facilitates functional recovery; where most of the literature that studies the effects of VR on the central nervous system describes changes in cortical circuits (and with them, also in motor performance) if appropriate virtual stimuli are presented to promote the adaptation of users’ motor patterns. This adaptation can be produced by the different elements of VR systems: the simplest ones, through sensory feedback, and the most complete or more immersive systems, facilitating learning based on practice in a motivating environment or learning based on the observation of reality action and/or imitation in a modified environment (being the last one that has shown the most changes in the activation of neural networks) [9]. 

Therefore, the topic of studying the benefits of VR in patients with AMI is a topic that remains open, subject to research that allows for the continued construction of knowledge; knowledge that the majority agrees that these types of treatments are safe and useful for patients given their playful aspect [11].

According to the scientific evidence in the literature, a direct relationship is detected between the improvements obtained in the different variables studied and the task time or its performance; a reduction of up to 22% in the performance of these tasks, and improvements in its functionality for patients who undergo physiotherapeutic treatment with VR as a complement [13].

It is visualized that, as a complement, it can be a strategy that could benefit the adherence and effectiveness of therapeutic treatments, since it will not only involve physical mechanisms that force the patient to practice musculoskeletal mobility, but it can also generate psychological well-being by promoting physical performance and physiological benefits through the implementation of this new physiotherapist alternative [9].

In this context, this paper aims to answer the research question: Does VR improve the speed of recovery and therapeutic adherence in the treatment for arthrogenic muscle inhibition?

In this paper, an investigation was carried out that answers the question posed, through a pilot test with 31 patients diagnosed with AMI subjected to rehabilitation, included VR. The development presented is related to Advances in Biomedical Sensing, Instrumentation and Systems. It integrates the measurement of biomedical signals from biomechanical instrumentation such as surface electromyography and dynamometry with the qualitative measurement of other variables like pain, functionality, and stiffness. This integration provides real-time comprehensive monitoring of the patient’s rehabilitation process. It is an example of the use of advanced biomedical sensing and instrumentation, combined with modern techniques such as biofeedback using virtual reality, to shift from the traditional subjective perception of the patient to a quantified measurement, enabling healthcare professionals to make more informed decisions. We describe an innovative approach to the acquisition of biomedical-related signals, their enabling technologies, and the interpretation of the data. Therefore, it focuses on the process of integrating technologies from diverse measurement systems, both qualitative and quantitative, in a biomedical application, and it is specifically related to bio-signal acquisition and biomedical sensing topics. Section 2 presents the materials and methods, and Section 3 presents the results considering the demographic and outcome variables of the research. Subsequently, in Section 4, the discussion of the results is presented. Finally, in Section 5, we present the conclusions.

## 2. Materials and Methods

All patients underwent a physiotherapeutic assessment by professionally trained personnel through tests such as: physical examination, walking or short walk, baropodometry, surface electromyography (sEMG), dynamometry, and jump tests, among others. The physiotherapist determined, according to the initial evaluation, several physiotherapy sessions that seek the well-being and improvement of the patient. After this, information is provided on the physiotherapy activities to be carried out, the inconveniences that may arise from the physiotherapy practice, and the duration and periodicity of the sessions; this through the socialization of the certificate of informed consent.

### 2.1. Inclusion Criteria

Patients declared with the biological condition of AMI as a product of a knee trauma.Healing process completed.Patients who can perform knee extension with at least 8 pounds of weight.Having a diagnosis of any of the following knee pathologies: injury or rupture of the anterior cruciate ligament (ACL), osteoarthritis, patellar tendinopathy, chondromalacia, mechanical knee dysfunction.Have at least 30 extra minutes after the end of the physiotherapy sessions.

### 2.2. Exclusion Criteria

Possessing some mental pathology.Being under the influence of any psychoactive substance.People who cannot have good visualization with a VR headset.

These exclusion criteria were adopted with the aim of avoiding potentially yielding false positives in the physiotherapist–patient relationship. While the exercise and measurement processes remained the same, adherence could be affected by the need for more structured communication, resources, or similar requirements according to health guidelines. The primary objective of the study was to compare conventional physical therapy versus virtual reality as individual methods to determine which one improved adherence more effectively, thus necessitating the removal of potential confounding factors.

### 2.3. Experimental Design

According to the statistics of the Exercise and Physiotherapy Center—Arthros in Medellin, Colombia—an average of 60 patients are received daily, of which half have pathologies associated with the knee, and of these 30 patients, around 10–12 present AMI.

To estimate the sample size, the parameter used was the change between the initial assessment and the final assessment of the dynamometry variable with both legs in a pilot test carried out with 5 people in the CG and 5 people in the IG (see Table 1). The calculation was made with a confidence level of 95%, a power of 80 and a control-intervention ratio of 1:1, and a size of 15 people was obtained for each group. 

This research is classified as an investigation with minimal risk, which has the endorsement of the Universidad Pontificia Bolivariana Ethics Committee (See Institutional Review Board Statement), since during the investigation a data record was obtained through procedures such as the use of two sEMG (mDurance, Granada-Spain) for the measurement of muscular electrical activity and an isometric dynamometer (Chronojump-Boscosystem, Barcelona-Spain) to obtain data on the force achieved during the exercises.

CG: Patients who received conventional therapy as recommended by the physiotherapeutic specialist.

IG: Patients who received conventional therapy as recommended by the physiotherapeutic specialist plus VR.

The study employed a sampling wherein groups were formed based on the accessible population within the institution. Initially, patients were randomly assigned to either the Control Group (CG) or Intervention Group (IG). However, due to differences in assessment durations, with CG patients typically taking 18–20 min and IG patients 22–30 min, IG patients indicating they had less than 30 min available were included in the CG. Despite this consideration for some time availability in the sampling, the subsequent analysis of demographic variables presented in Section 3.1 (Demographic characteristics subsection) revealed that these variables did not exhibit statistically significant differences affecting the study.

Next, three assessment dates were agreed upon: the initial assessment, at the beginning of the physiotherapy treatment, the intermediate assessment in the intermediate treatment session and the final assessment in the last physiotherapy session. If at the time of carrying out the intermediate assessment the patient manifested a lack of availability, this assessment could be carried out in any of the following two sessions.

In the sessions where it had been agreed to make an assessment, once the session was over, the surface electrodes were placed on the muscles to be assessed: vastus medialis and vastus lateralis, which contribute to the extension and stability of the knee in daily activities and sports performance [14].

If the patient belonged to the CG and had finished the physiotherapy session, the electrodes and sEMG were placed on them, according to the project Surface Electromyography for the non-invasive assessment of muscles (SENIAM) [15] and they were asked to use the knee extension machine connected to the load cell of the Chronojump brand dynamometer for an assessment of strength. Once the patient was seated, we proceeded with the practical explanation of how the execution would be, which consisted of performing maximum force (RM) in knee extension, with the foot in a neutral position until there was no discomfort or pain in the knee or at least that it was tolerable and that it did not compromise the patient’s pathology. The exercise was initially conducted bilaterally and then unilaterally, each for a maximum time of 6–7 s with rest intervals of 90 s [16] between each execution, so that the patient could recover energy after the energy expenditure during the exercise test [17].

If the patient belonged to the IG and had finished the physiotherapy session, the electrodes and sEMG were placed on them, according to the project SENIAM [15], they put on the VR headset and was placed in a chair with a space of approximately 15 m of free access, so that the patient could visualize, replicate, and execute all the movements simultaneously with the character in the video (go to Appendix A) for 7–8 min. For this purpose, there were 2 types of videos, one for women and another for men according to the sex of the patient. The videos were filmed from a first- and third-person perspective as seen in Table 2, featuring the physiotherapist correctly performing the exercises to allow patients to replicate them while watching. 

This content is presented as an immersive VR experience through VR glasses branded as VR 3D BOX (VR BOX VIRTUAL REALITY GLASSES Dongguan-China). These glasses offer a field of view between 85°–95° and are compatible with mobile device screens ranging from 3.5″ to 6″. They provide freedom for the patient’s execution and mobility [18].

After viewing the video, the patient was given 90 s of recovery [16] and they were invited to go to the knee extension machine connected to the load cell of the Chronojump brand dynamometer to perform a strength assessment. Once the patient was seated, we proceeded with the practical explanation of how the execution would be, which consisted of performing the RM in knee extension, with the foot in the neutral position until there was no discomfort or pain, or at least that it was tolerable and that it did not compromise the patient’s pathology. The exercise would initially be conducted bilaterally and then unilaterally, each for a maximum time of 6–7 s with rest intervals of 90 s [16] between each execution so that the patient could recover energy after the energy expenditure during the test [17].

A process summary including the VR experience is shown in Figure 1, and some images of the VR environment in participants using the VR experience are presented in Table 2.

Note that the procedure for the acquisition of the force signal for both groups is the same and while it was being conducted, dynamometric and sEMG data were captured. In the initial assessment of each patient, they filled out two questionnaires, a sports activity questionnaire (See Appendix A) and a Western Ontario and McMaster Universities Osteoarthritis (WOMAC) Index symptom questionnaire (See Appendix A) [19], while in the intermediate and final assessment they only filled out the WOMAC symptom questionnaire.

### 2.4. Outcome Variables

The outcome variables in the study are the following:

According to the sport injury rehabilitation adherence scale (SIRAS) [12], it is possible to calculate therapeutic adherence under the parameters of intensity, frequency to follow instructions and advice and receptivity of the patient during the session, with a scale from [0–100%] for each item. In this paper, allusion will be made to therapeutic adherence such as attendance, willingness, and receptivity in the sessions; items assessed through the SIRAS scale, where willingness and receptivity were assessed by the treating physiotherapists [20,21,22].

**Therapeutic adherence**: it was calculated as the average of three variables: attendance, disposition, and receptivity to physiotherapy sessions.

*Attendance:* for each patient, this was calculated through Equation (1):(1)$$att=\frac{ass_{s}}{sch_{s}}\times 100\%$$
where $ass_{s}$ are assisted sessions, those in which the patient was present and fully completed their rehabilitation session and $sch_{s}$ are scheduled sessions, those scheduled by the Arthros Center for each patient.

*Disposition*: it was assessed by the treating physiotherapist, who determined with a percentage value, on a scale of [0–100%], the attitude with which each patient attended their rehabilitation process [8], especially taking into consideration the assessment sessions (initial, intermediate, final).

*Receptivity*: it was evaluated by the treating physiotherapist, who defined it with a percentage value, on a scale of [0–100%], the way in which the instructions of the physiotherapist professionals were received by the patients [8], in the same way with the main emphasis on the assessment sessions.

**Recovery speed** is the weighting of the following variables:

*Pain level:* the WOMAC questionnaire [19] was used, (See Appendix A), which has 6 questions to assess the level of pain. Each question has a range of [0–4]. The responses of each patient were added, with which there was a possible range per patient from 0 to 24. The level of pain was normalized as Equation (2):(2)$${Painnorm}_{n}=\frac{{Painpat}_{n}}{VmaxPain}$$
where ${Painnorm}_{n}$ is the total normalized pain patient in the assessment n and corresponds to the sum of the 6 questions for each patient in the assessment n (${Painpat}_{n}$) divided by VmaxPain that corresponds to the value obtained by the patient with the highest sum (including the three assessments of the two groups).

*Stiffness level:* it was evaluated through the WOMAC questionnaire [19], which can be seen in Appendix A, and it has 3 questions on the level of stiffness. Similarly, it has a response range of [0–4]. Responses from each patient were added, giving a possible range per patient of [0–12]. The stiffness level was normalized as Equation (3):(3)$${Stiffnorm}_{n}=\frac{{Stiffpat}_{n}}{VmaxStiff}$$
where $Stiffpat_{n}$ corresponds to the total stiffness assessment in the session n and it is the sum of the 3 questions of each patient in this assessment, and VmaxStiff corresponds to the value obtained by the patient with the highest sum (including the three assessments in the two groups).

*Difficulty level:* it was evaluated using the WOMAC questionnaire [19], and it has 20 questions about the level of difficulty in the execution of basic and daily activities of the patients in previous days to the assessment session. Each answer has a value of [0–4], and therefore the maximum possible score is 80. The level of difficulty was normalized as Equation (4):(4)$${Diffnorm}_{n}=\frac{{Diffpat}_{n}}{VmaxDiff}$$
where ${Diffpat}_{n}$ corresponds to the sum of the 20 questions of each patient in each assessment and $Vma{xDiff}_{}$ corresponds to the value obtained by the patient with the highest sum (including the three assessments in the two groups).

*Dynamometry or force measurement:* the data captured through the Chronojump brand dynamometer [23] were standardized by the relationship between maximum force and body weight so that the relative force was obtained [24] as Equation (5):(5)$${Frelpat}_{n}=\frac{Fmax{abs}_{n}}{m_{n}g}$$
where ${Fmaxabs}_{n}$ corresponds to the maximum value captured with the dynamometer in the isometric knee extension–flexion machine, and $m_{n}g$ [Kg] corresponds to the value delivered by the scale for each of the patients. The max normalized relative force was calculated as Equation (6):(6)$${Frelnorm}_{n}=\frac{{Frelpat}_{n}}{VmaxFrel}$$
where ${Frelpat}_{n}$ is the force value obtained in the knee flexion–extension test normalized with (5) and VmaxFrel corresponds to the highest value of ${Frelpat}_{n}$ that was obtained in the three assessments in the two groups after standardization. 

*Muscle electrical activity:* this corresponds to the sEMG signal acquired through an mDurance sEMG equipment. This variable was acquired, and these data were normalized [25]. The measurement of the maximum voluntary contraction (MVC) of the vastus lateralis and medialis [14] was taken when doing bilateral contraction and when doing individual contraction for each leg, for a total of 8 values: electrical activity vastus lateralis left leg knee flexion–extension with both legs, electrical activity vastus medialis left leg knee flexion–extension with both legs, electrical activity vastus lateralis right leg knee flexion–extension with both legs, electrical activity vastus medialis right leg knee flexion–extension with both legs, electrical activity vastus lateralis left leg knee flexion–extension with left leg, electrical activity vastus medialis left leg knee flexion–extension with left leg, electrical activity vastus lateralis right leg knee flexion–extension with right leg, and electrical activity vastus medialis right leg knee flexion–extension with right leg.

Subsequently, standardization was made for each value with the repetition maximum (RM). That is, each MVC of each muscle will be divided by the RM to be able to compare between users/patients with different physical characteristics, compare between muscles of different sizes and types of fibers, and compare the state of a patient at different stages of rehabilitation or training [26]. Muscle electrical activity was standardized as Equation (7):(7)$${MVCstand}_{n}=\frac{{MVCpat}_{n}}{1RM}$$
where $MVCpat_{n}$ corresponds to the electromyographic peak, in µV, of each vastus lateralis and medialis of both extremities captured with surface electromyographs, and 1RM corresponds to the maximum weight in kilograms that the patient was able to move for a single repetition, in this case, a repetition of flexor isometric knee extension. The standardized muscular electrical activity was normalized as Equation (8):(8)$${MVCnorm}_{n}=\frac{{MVCstand}_{n}}{VmaxMVC}$$
where $MVC\_stand_{n}$ corresponds to the standardized electromyographic value of each patient in each assessment with (7), and VmaxMVC corresponds to the highest value of the $MVCstand_{n}$ that was obtained in the three assessments in the two groups after standardization.

For the weighting of each of the variables that make up the speed of recovery, a questionnaire was administered to 14 physiotherapists where each one quantified [0–100%] the importance of the variables, which can be seen in Appendix A—Importancia de las variables de desenlace and Table 3, according to their academic, professional criteria and experience, in a way that allows them to determine when a patient is recovered. It was a sampling considering the physiotherapist population accessible in the institution.

With this, an importance value of each item was generated for each physiotherapist, a value that was normalized as Equation (9):(9)$${Wvarnorm}_{i_{f}}=\frac{{Wvar}_{i}}{\sum_{i=1}^{5}{=Wvar}_{i_{f}}}$$
where ${Wvar}_{i}$ [0–100%] is the weight of the variable *i* for the physiotherapist *f*, according to Table 3, and the denominator of the equation is also a value in the range [0–100%].

Finally, the variable importance level was calculated as an average of each normalized variable and was obtained as Equation (10):(10)$${IL}_{i}=\frac{\sum_{f=1}^{14}{Wvarnorm}_{i_{f}}}{14}$$
where i was replaced for each case by the name of the variable to be weighted according to Table 3.

Thus, with all the normalized and dimensionless variables, each of the variables multiplied by the level of importance value was summed up and thus the speed of recovery of each patient in each evaluation was obtained. It is highlighted that, for the variables of pain, stiffness, and difficulty, the lower the value, the better the indicator, while for strength and muscular electrical activity it is better that the value be higher; therefore, the recovery speed was determined according to Equation (11):(11)$$\begin{array}{c} {Recspeed}_{n}=\left(1-{Painnorm}_{n}\right)\times IL_{Pain}+\left(1-{Stiffnorm}_{n}\right)\times IL_{Stiff}+\left(1-{Diffnorm}_{n}\right)\times IL_{Diff}\\ +({Frelnorm}_{n})\times {IL}_{F}+({MVCnorm}_{n})\times {IL}_{MVC} \end{array}$$

In addition to this, for dynamometry and muscular electrical activity, the symmetry of the legs of each patient was calculated.

*Symmetry of force*: it was used to determine the evolution and the relationship between the extremities of each user and therefore of each group. The force symmetry was calculated as Equation (12):(12)$${Fsym}_{n}=\frac{{VminF}_{n}}{{VmaxF}_{n}}$$
where ${VminF}_{n}$ is the minimum value of force between the left leg and the right leg in the same attempt and ${VmaxF}_{n}$ is the maximum value of force between the left leg and the right leg in the same attempt.

*Symmetry of muscular electrical activity:* the symmetry of electrical activity was calculated as Equation (13):(13)$${MVCsym}_{n}=\frac{{VminMVC}_{n}}{{VmaxVMC}_{n}}$$
where ${Vmin\_MVC}_{n}$ is the minimum EMG peak value of the vastus lateralis or medialis on the same attempt, and ${Vmax\_VMC}_{n}$ is the maximum EMG peak value of the right or left leg vastus lateralis or medialis in the same attempt.

### 2.5. Analysis Plan

Due to the small size of the sample, non-parametric statistics were used to analyze the results obtained.

The data of the 3 evaluations carried out on each patient, in their respective groups (CG or IG), were analyzed, and it was compared which group achieved a greater speed of recovery and a better therapeutic adherence during the physiotherapy treatment. 

Preliminarily, the following tests were defined with demographic information and outcome variables:The Mann–Whitney U test was used to establish differences in numerical demographic variables such as age, weight, height, body mass index (BMI), average hours trained per week, resistance activities performed before the pathology, strength activities that they performed before the pathology, balance activities that they performed before the pathology and flexibility activities that they performed before the pathology for the CG and the IG. They are considered statistically significant if the value *p* < 0.05.The chi-square test was used to assess the association of non-numerical demographic variables such as gender, affected limb (both, right, left) and pathology for the CG and the IG. They are considered statistically significant if the value *p* < 0.05. *p* is expected to be greater than 0.05.The Friedman test was used to evaluate the change of the different parameters measured in the three moments, some of these were: therapeutic adherence, strength with both legs (standardized and normalized), strength with the left leg (standardized and normalized), strength with the right leg (standardized and normalized), muscle electrical activity, force symmetry, and electrical activity symmetry. They are considered statistically significant if the value *p* < 0.05.A delta was estimated (final score—initial score) for each of the outcome variables and the Mann–Whitney U test, Student’s t test, and chi-square were used to assess the delta differences between the two groups.

Data for quantitative variables are presented in medians, 25th, 50th and 75th percentiles or mean and standard deviation depending on normality and qualitative variables as absolute frequencies and percentages.

## 3. Results

### 3.1. Demographic Characteristics

The variables age, weight, BMI, resistance activities before the pathology, people with both limbs affected and people with the right leg affected have a slightly higher value for the IG compared to the CG; however, there is no significant difference. This means that the sample was adequately distributed according to the demographic variables and that the results obtained are typical of the intervention carried out. Similarly, there was no significant difference in the population by pathology or conditions associated with the knee as seen in Table 4.

From Table 4, it is possible to determine that the demographic variables do not have a statistically significant difference that affects the study, because the *p*-value in each of the variables is greater than 0.05; being able to affirm that the distribution of the patients to the CG and the IG was homogeneous. Additionally, from the same table, it is possible to affirm that:

The CG population was younger compared to the IG; however, there was no statistically significant difference between the two groups.The weight of the CG was less than the weight of the patients in the IG.Considering that the CG had fifteen patients and the IG had sixteen, the distribution of patients regarding gender was uniform, since there were eight and seven male patients for the CG and IG, respectively.Height was homogeneous between the two groups, and although the CG was slightly taller than the IG, the difference between the 25th and 75th percentiles is not significant.Body mass index (BMI) was slightly higher in the IG compared to the CG; however, the difference is not statistically significant to influence the research project.Regarding the average number of hours that the patients in the CG and IG train per week and the type of sporting activity they perform, be it resistance, strength, balance, or flexibility; it is evident that they are not differentiating factors that impact the research project and the variables in Table 4, thus again affirming that there was a homogeneous distribution of the population.Regarding the affected limb, three patients with both legs belong to the CG while four belong to the IG; six people with pathology in the right leg for the CG and ten for the IG; and the CG had six patients with the affected left leg and the IG had two.Regarding the pathology, the *p* value between the CG and IG is greater than 0.05, therefore, there is no statistically significant difference between the two groups and the 19 different ailments.

### 3.2. Outcome Variables

Table 5 shows the results of the statistical analysis performed on each of the variables measured by the patients in the CG and the IG.

## 4. Discussion

In the present intervention and analytical study, it was found that Virtual Reality (VR) generates important benefits in therapeutic adherence and in the recovery of the Arthrogenic Muscular Inhibition (AMI) of the knee, improving the functional capacity of the patient, evidenced in the reduction of pain, increase in the strength in both legs and symmetry of the same, and muscular electrical activity, mainly of the vastus lateralis. The foregoing is supported by the following relevant aspects:Improvement in the therapeutic adherence between the Control Group (CG) and Intervention Group (IG).Increased strength with both legs and symmetry between them.Decreased level of pain and stiffness that is related to mobility.

The main results will be shown in the Table 6:

### 4.1. Improvement in the Therapeutic Adherence between CG and IG

According to [1], in many cases AMI hinders rehabilitation after knee joint injury by preventing functional activation of the quadriceps. This has been attributed to neural reflex activity originating in the injured joint and producing decreased efferent drive to muscles, such as the quadriceps, therefore rehabilitation protocols after knee injury should focus in neuromuscular and mechanical alterations, improving therapeutic adherence more than in the pathology itself, since in most cases AMI is chronic. Our study shows an improvement in adherence, for the control group, therapeutic adherence begins with a median of 91.7 and then goes to 93.3, which indicates an increase in the adherence score in the 33.3% of patients, with a *p* value of 0.022, which means that there is a statistically significant difference, especially between the initial assessment and the final assessment. For the IG, adherence starts with a median of 94.2 and goes to 96.7, an increase of 37.5%, statistically significant with a *p* value of 0.007, indicating a significant difference and a change in the initial assessment with respect to the final assessment for this cluster.

### 4.2. Increased Strength with Both Legs and Symmetry between Them

Regarding the normalized force in knee flexion and extension with both legs MVIC in the CG, there was a statistically significant difference since the *p* value is 0.013, an initial median of 8.26, intermediate of 9.41, and final of 10.7 for a delta of 1.22 between the final and initial assessment; while for the IG the delta between the final assessment, with a median of 9.40, and the initial one with a median of 7.83, was 1.25, being slightly higher for the IG. This is in line with the study by Bartholdy in [1], in which he found a statistically significant difference in favor of American College of Sport Medicine interventions with respect to knee extensor strength (SMD difference: 0.448 (CI 95%: 0.091–0.805)). In the Bartholdy study [1], no non-American College of Sport Medicine interventions were observed and associations between increased knee extensor strength and changes in pain and/or disability were assessed using meta-regressions that indicated that increases in knee extensor strength of 30–40% would be necessary for a likely concomitant beneficial effect on pain and disability, respectively.

Regarding the symmetry of the force in flexion–extension with the left leg and the right leg normalized MVIC, an increase and significant difference between the initial assessment with a median of 84.5 and the final assessment with a median of 91.4 is demonstrated within the IG and between the intermediate assessment with a median of 90.9 and the final assessment. While for the CG there was no significant difference within the group.

Previous studies have shown that electrical stimulation of the common peroneal nerve concurrent with maximum voluntary effort can eliminate AMI of the quadriceps in OA of the knee, being beneficial for rehabilitation; on the other hand, virtual reality has also been shown as an innovative process for the recovery of motor function in the rehabilitation of neurological patients [27], and in the re-education of perceptual deficits. In our study we validated the benefits of electrical stimulation for lower limb musculoskeletal injuries. In the normal muscular electrical activity of the vastus medialis of the left leg when the patients perform knee flexion–extension with both legs, there is no significant difference between the deltas of CG and IG, *p* value of 0.711. This is extrapolated from other studies in this regard and is also consistent with studies by Harkey [1], in which neuromuscular electrical stimulation produced weak negative to strong positive effects (value range 1/4 0.50 to 1.7) over a period of 3 weeks to 6 months.

When analyzing the normalized muscular electrical activity of the vastus medialis of the right leg when the patient performs knee flexion–extension with both legs and in particular the change inside the CG, it is determined that there is a negative change between the final assessment and the initial assessment of the two groups. Regarding the normalized muscular electrical activity of the vastus lateralis of the right leg with knee flexion–extension, there is an increase in the median from the initial assessment to the final assessment in the CG and the IG.

Now, for the vastus lateralis of the left leg when the patient performs knee flexion–extension with the left leg, the IG achieved a greater difference between the final assessment and the initial assessment with respect to the same assessments of the CG; this being positive for the IG. Now, when verifying the comparison between couples within each group, the CG did not have a significant difference, with a *p* value of 0.549, and the IG did not have a significant difference either, with a *p* value of 0.305; however, the change between the median of the final assessment and the initial assessment of the GI is greater than that of the CG.

When analyzing the results of the symmetry of the normalized muscular electrical activity of the vastus lateralis when the patients flexed and extended the knee with both legs between the CG and IG, it was found that while the CG had no statistically significant changes, with a *p* value of 0.863, from the initial assessment with a median of 86.4, the intermediate assessment with 74.8 and the final assessment with 78.6; the IG had a *p* value of 0.006 with significant changes, especially between the intermediate assessment with a median of 61.5 and the final assessment with 82.7. This result is positive for the IG and would generate an improvement in the symmetry between the patient’s extremities.

### 4.3. Decreased Level of Pain and Stiffness That is Related to Mobility

Regarding the level of pain for the CG, the median with a value of 3 in the initial assessment decreased to 2 in the intermediate and final assessment. While for the IG, the initial assessment median of 4.5 went to 4 in the intermediate assessment.

When individually analyzing the level of stiffness of each group, it is found that the *p* value of the CG is 0.005, which is statistically significant; and it can be seen in the change in the level of stiffness from the initial assessment with a median of 2, then it goes to 1 in the intermediate assessment and ends with a median of 0 and a 75th percentile of 1.5 in the final assessment; for the IG, the level of stiffness decreases from the initial assessment with a median of 2.5 to 1 in the final assessment; thus, it is a good indicator of the decrease in the level of stiffness of the patients in this group.

In addition to the main findings in terms of adherence, strength, and mobility difficulty, some additional analyses were performed:

Injuries of musculoskeletal origin are considered a public health problem, being a frequent cause of physiotherapeutic consultations, occupying 80% of the patients who attend this service [1], and of this, 30% are young, the remaining adult population being older. The CG population was younger compared to the IG; however, there was no statistically significant difference between the two groups; it should be clarified that the origin of the AMI was variable, in one the discharge of the mechanoreceptors of the affected joint (CG) is increased and on the other hand, in structural lesions of the joint receptors, such as osteoarthritis or in traumatic injuries, it is decreased with the afferent discharge of these (IG). At the time of the study, the young population had an AMI where, in medical terms, recovery could take 10 to 15 days, while in the intervention population, that is, adults, it took months and years, and this occurred in a limited window of time. The IG received additional VR treatment compared to the CG, thus the efficacy of the treatment may have been due to the additional time provided for the IG, making it important to further study the potential effects of virtual reality.

For the present work, it is important to highlight that the Arthros Center in its day-to-day exercise involves techniques that support pain reduction, elimination and treatment of inflammation and edema, strengthening work, motor control, proprioceptive exercises (stability), sports rehabilitation, and PEACE & LOVE methodology. Considering the above, it was compared with a CG in which the internal study variables were already favored by the rehabilitation strategies, which is positive, because it indicates that the results should be even higher in a comparative way if the CG corresponded to users of traditional rehabilitation.

Most of the reviewed publications found diverse effects in the approach specifically in the knee; possibly because this joint has been widely studied due to its opposite mechanics: mobility to allow movement of the body and joint movements, and stability to support body weight, as well as the load in the stance phase of walking. However, it is understandable that joint damage can cause AMI in any joint, affecting the functionality of the individual, which may lead to a greater field of study in other joints of the human body. Although the appearance of AMI cannot be avoided, it can be treated, and its severity reduced with appropriate physiotherapeutic intervention. In this sense, it is possible to promote the recovery of the patient from the perspective of conventional physiotherapy, by simulating tasks like activities of daily living with a greater probability of benefits due to the repetitive, intensive, and emotional intervention provided by VR.

Some knowledge-related implications that can be highlighted from this studio include a deeper understanding of AMI as a condition that can lead to biomechanical motor control and movement problems, as well as muscle weakness and atrophy. Furthermore, the study underscores the integration of VR methods into physiotherapy practices, showing that this technology can be well-perceived by physiotherapists and has the potential to enhance treatment outcomes. The study also highlights the use of dynamometry and surface electromyography (sEMG) signal acquisition tools for assessing the effects of VR in physiotherapy, emphasizing the importance of utilizing advanced measurement techniques in clinical research.

From a theoretical perspective, virtual reality could serve as a starting point for future research in neurological conditions among more vulnerable populations such as the blind, amputees, individuals with dementia, or those with neurocognitive disorders, where perception and proprioception of movement execution are limited. 

Some practical implications include the finding that the research suggests incorporating VR as a complement to traditional treatment significantly improved therapeutic adherence. This implies that patients may become more engaged and committed to their rehabilitation when VR is part of the therapy. Also, the study indicates that the use of VR may lead to a faster recovery process for individuals with AMI, especially increased strength in both legs and improved symmetry between them. This practical implication is beneficial for physiotherapists, as it suggests that VR can contribute to better physical outcomes for patients. Moreover, the research demonstrated a decreased level of pain and stiffness related to mobility in patients who used VR as part of their treatment. This suggests that VR may assist in pain management and improve mobility in clinical settings.

In summary, healthcare providers and institutions may consider integrating VR into their physiotherapy programs, especially for patients with conditions like AMI, to potentially enhance treatment outcomes, patient satisfaction, and adherence. In practice, there is still a need for more resources and additional training for various healthcare professionals, including doctors, nurses, physiotherapists, and physiatrists in this field. However, it is hopeful to consider it as a potential within the spectrum of palliative and curative medical approaches in rehabilitation.

The estimation of the sample size made a difference of 95 in the dynamometry varia-tion [N]; however, in the study measurements when comparing the two groups, no differences of this size were found, which suggests the need to increase the sample sizes.

## 5. Conclusions

After executing this research, the conclusion and answer to the question posed initially, “Does VR improve recovery speed and adherence in the treatment for arthrogenic muscle inhibition?” and, taking into account that therapeutic adherence is the weighting of the variables attendance, willingness, and receptivity to physiotherapy sessions, and the speed of recovery as the sum of the variables decrease in the level of pain, decrease in the level of stiffness, decrease in the level of difficulty, increase in strength and increase in muscle electrical activity, is that there is not a statistically significant difference between the patients in the CG and the IG undergoing the VR physiotherapy treatment, since the patients improved in a similar way. However, analyzing the variables individually, it is possible to affirm that virtual reality (VR), as a complement, statistically significantly improves therapeutic adherence, the normalized force in knee flexion–extension with both legs MVIC, the normalized force symmetry between knee flexion–extension individual with left leg and right leg, normalized muscular electrical activity of the vastus lateralis of the left leg in knee flexion–extension with both legs, normalized muscular electrical activity of the vastus lateralis of the left leg with knee flexion–extension with the left leg, normalized muscular electrical activity of the vastus lateralis of the right leg in knee flexion–extension with the right leg, symmetry of normalized muscular electrical activity of the vastus lateralis in knee flexion–extension with both legs, symmetry of the normalized muscular electrical activity of the vastus lateralis in individual knee flexion–extension, and the decrease in the level of stiffness and the level of difficulty in carrying out basic and daily activities. This can be attributed to the factor of adherence to the treatment that VR generates, since the patient is submerged in an environment that motivates them to replicate and try to execute the movements that they visualize in the VR video extensively, that is, with a wider range of motion; therefore, the level of stiffness decreases, functionality increases and this means that the patient can recruit more muscle fibers to perform knee flexion–extension, which helps to increase the excitability of the muscles surrounding the knee and consequently increase muscle electrical activity, thus the greater the recruitment of muscle fibers, the greater the force.

This research has the potential to be applied to fields such as physiotherapy or topics related to the musculoskeletal system’s movement. The findings of this research provide a foundation for the development of other wearable, miniaturized, and portable technological systems that could even enable physiotherapy professionals to benefit from patients located in environments outside the clinic.

Future works include expanding the sample size for both groups, the CG and IG, which will allow significant differences to be detected. Additionally, it is interesting to consider a comparison between VR treatment and traditional physical therapy, since the CG in this study already includes some innovative elements in its treatment technique. In addition, it is interesting to explore other VR techniques that can evaluate the progress in the rehabilitation process. 

## Figures and Tables

**Figure 1 sensors-23-09114-f001:**
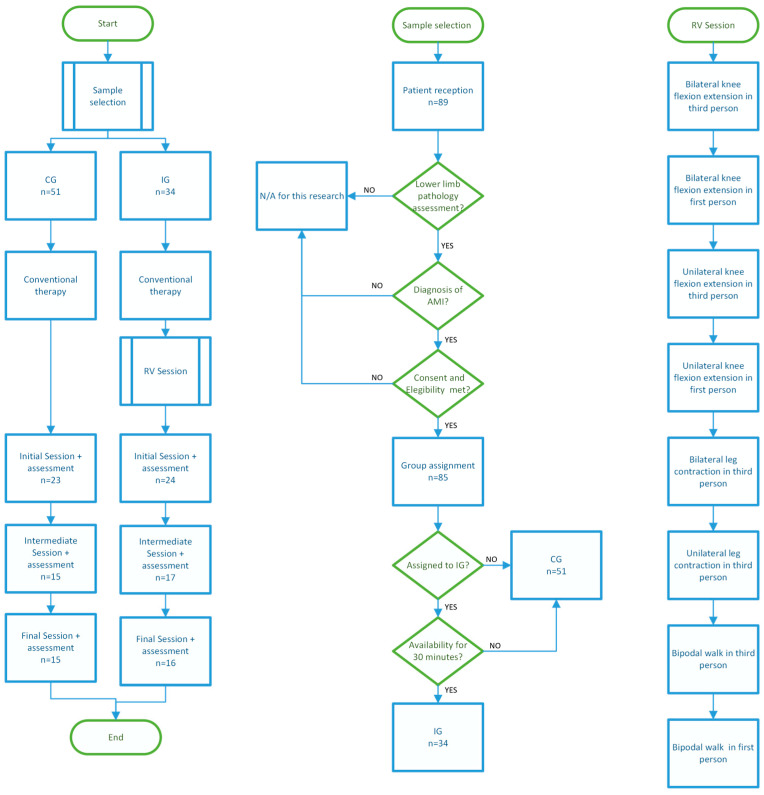
Methodology flowchart.

**Table 1 sensors-23-09114-t001:** Pilot test for test estimation.

Dynamometry with Both Legs [N]						
	Mean of Force [N]	Standard Deviation of Force [N]	Initial Assessment	Intermediate Assessment	Final Assessment	Difference Final Assessment–Initial Assessment
CG	165.33	55.10				
1			905.35	993.82	1007.14	101.79
2			437.96	500.72	586.77	148.81
3			166.4	177.43	358.37	191.97
4			981.08	1058.92	1119.75	138.67
5			280.82	480.68	526.24	245.42
IG	85.46	71.61				
1			287.17	344.1	315.19	28.02
2			572.7	683.3	736.32	163.62
3			599.97	644.25	749.59	149.62
4			364.8	344.53	442.62	77.82
5			632.76	636.12	624.56	8.2

**Table 2 sensors-23-09114-t002:** Images from the VR videos.

Symbol	Activity	VR Environment Images		Patients Using VR
		Women	Men	
BKFE3	Bilateral knee flexion–extension in third person	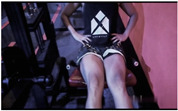	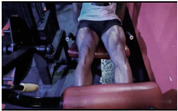	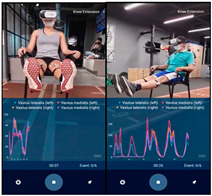
BKFE1	Bilateral knee flexion–extension in first person	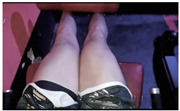	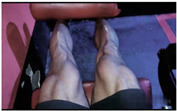	
UKFE3	Unilateral knee flexion–extension in third person	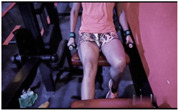	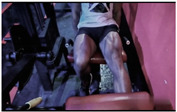	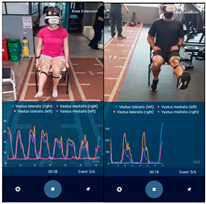
UKFE1	Unilateral knee flexion–extension in first person	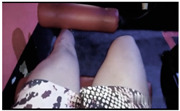	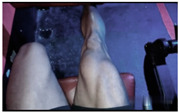	
BLC3	Bilateral leg contraction in third person	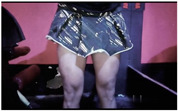	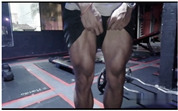	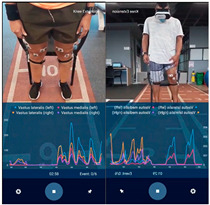
UC3	Unilateral leg contraction in third person	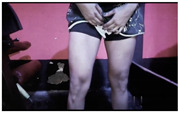	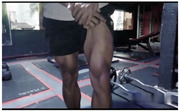	
BW3	Bipodal walk in third person	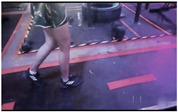	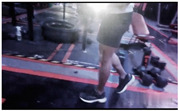	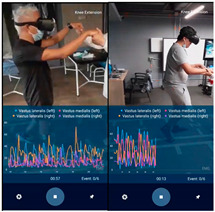
BW1	Bipodal walk in first person	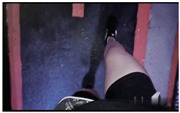	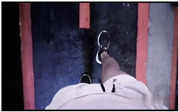	

**Table 3 sensors-23-09114-t003:** Level of importance of the variables that make up the speed of recovery.

Variable	Normalized Level of Importance
Pain (Pain)	0.21
Stiffness (Stiff)	0.22
Difficulty (Diff)	0.22
Force (F)	0.20
Muscle electrical activity (MVC)	0.15

**Table 4 sensors-23-09114-t004:** Demographic characteristics.

Characteristic	CG N = 15 Median (IQR); *n* (%)	IG N = 16 Median (IQR); *n* (%)	*p*-Value
**Age**	29 (22.5, 36.5)	33.5 (29.3, 52.3)	0.10
**Weight**	63 (57.9, 69)	68 (59.5, 76.3)	0.43
**Gender (Male)**	8 (53%)	7 (44%)	0.59
**Height**	1.73 (1.65, 1.75)	1.70 (1.64, 1.74)	0.65
**BMI**	22.2 (20.7, 22.7)	23.7 (21.2, 26.5)	0.16
**Average weekly training hours**	8 (3, 13)	8 (3, 13)	>0.99
**Number of resistance activities before pathology**	5 (4, 6)	5.5 (3.75, 8)	0.60
**Number of strength activities before pathology**	4 (3, 5)	3.5 (2, 4)	0.67
**Number of balance activities before pathology**	2 (2, 3)	2 (1, 3)	0.55
**Number of flexibility activities before pathology**	0 (0, 0)	0 (0, 0.250)	0.65
**Affected limb**			0.26
Both	3 (20%)	4 (25%)	
Right	6 (40%)	10 (62.5%)	
Left	6 (40%)	2 (12%)	
**Pathology**			0.60
Osteoarthritis	0 (0%)	1 (6.2%)	
Chondromalacia	3 (20%)	1 (6.2%)	
Knee bruise	0 (0%)	1 (6.2%)	
Meniscus tear	0 (0%)	1 (6.2%)	
Cartilage tear, right knee osteotomy	1 (6.7%)	0 (0%)	
Patella deviation	0 (0%)	1 (6.2%)	
Mechanical knee dysfunction	1 (6.7%)	1 (6.2%)	
Knee sprain	0 (0%)	1 (6.2%)	
Symmetric bilateral genu varus with decreased medial intercondylar space + bilateral patellofemoral disorder + goose foot syndrome + right plantar fasciitis	1 (6.7%)	0 (0%)	
Genu varus knee	1 (6.7%)	0 (0%)	
Baker’s cyst	1 (6.7%)	0 (0%)	
Anterior cruciate ligament rupture	1 (6.7%)	2 (12%)	
Deviated patella + tendinitis	2 (13%)	0 (0%)	
Lateralized patella	1 (6.7%)	0 (0%)	
Iliotibial band syndrome	1 (6.7%)	0 (0%)	
Patellofemoral syndrome	0 (0%)	2 (12%)	
Patellofemoral syndrome + deviated patella	0 (0%)	1 (6.2%)	
Patellar tendinopathy	2 (13%)	3 (19%)	
Bilateral patellofemoral disorder	0 (0%)	1 (6.2%)	

**Table 5 sensors-23-09114-t005:** Outcome variables.

Characteristic	GC N = 15 Median (IQR)			*p*-Value	Comparisons between Pairs	Δ GC Mean (Standard Deviation); *n* (%) ^a^; Median (IQR)	GI N = 16 Median (IQR)			*p*-Value	Comparisons between Pairs	Δ GI Mean (Standard Deviation); *n* (%) ^a^; Median (IQR)	*p*-Value Δ GC vs. Δ GI
	**Initial assessment = V1**	**Intermediate assessment = V2**	**Final assessment = V3**				**Initial assessment = V1**	**Intermediate assessment = V2**	**Final assessment = V3**				
Therapeutic adherence	91.7 (89.2, 100)	93.3 (90, 100)	93.3 (90, 100)	0.022 ^b^	V1–V3	5 (33.3%)	94.2 (91.7, 100)	96,7 (91.7, 100)	98.3, (93.3, 100)	0.007 ^c^	V1–V3	6 (37.5%)	0.809 ^d^
Attendance	100 (100, 100)	100 (100, 100)	100 (100, 100)				100 (100, 100)	100 (100, 100)	100 (100, 100)				
Disposition	85 (80, 100)	85 (80, 100)	85 (80, 100)	0.050 ^e^	V1–V2 V1–V3	3 (20%)	100 (85, 100)	100 (90, 100)	100 (90, 100)	0.018 ^f^	V1–V3	5 (31.3%)	0.685 ^g^
Receptivity	90 (85, 100)	90 (85, 100)	95 (85, 100)	0.097		3 (20%)	100 (80, 100)	100 (83.8, 100)	100 (85, 100)	0.061 ^h^	V1–V3	3 (18.8%)	1 ^i^
Normalized knee flexion–extension force with both legs MVIC	8.26 (6.67, 11.2)	9.41 (8.18, 11.7)	10.7 (8.46, 11.8)	0.013 ^j^	V1–V2	1.22 (2)	7.83 (5.58, 8.98)	8.98 (6.10, 10.7)	9.40 (6.98, 11.3)	<0.001 ^k^	V1–V3 V2–V3	1.25 (1.34)	0.963 ^l^
Normalized knee flexion–extension force with left leg MVIC	3.86 (3.68, 5.95)	5.35 (4.22, 6.38)	5.22 (4.69, 6.22)	0.009 ^m^	V1–V2	0.835 (1.34)	4.06 (3.22, 4.78)	4.70 (3.33, 5.57)	5.07 (3.79, 5.82)	<0.001 ^n^	V1–V3 V2–V3	0.656 (0.534)	0.625 ^o^
Normalized knee flexion–extension force with right leg MVIC	4.74 (3.88, 6.08)	5.66 (4.40, 6.46)	5.42 (4.76, 6.63)	0.025 ^p^	V1–V2	0.826 (1.51)	4.80 (3.53, 5.74)	4.82 (3.76, 5.99)	5.23 (3.75, 6.14)	0.004 ^q^	V1–V3	0.439 (0.524)	0.340 ^r^
Symmetry of force between knee flexion–extension with normalized left and right leg MVIC	88.7 (85.5, 89.9)	89.5 (85.5, 97.4)	90 (86.9, 97.3)	0.420		4.94 (9.69)	84.5 (73.1, 96.4)	90.9 (81.6, 94.9)	91.4 (86.9, 98.8)	0.015 ^s^	V1–V3 V2–V3	8.63 (9.30)	0.287 ^t^
Normalized muscular electrical activity vastus lateralis left leg knee flexion–extension with both legs	6.87 (4.80, 8.73)	5.04 (4.17, 9.05)	7.08 (5.57, 8.78)	0.155		2.22 (−0.662, 3.06)	6.36 (4.52, 11.6)	6.31 (3.78, 10.3)	8.43 (4.81, 11.1)	0.444		1.29 (−1.72, 3.59)	0.770 ^u^
Normalized muscular electrical activity vastus medialis left leg knee flexion–extension with both legs	6.20 (3.75, 10.8)	6.20 (3.85, 7.42)	5.0 (4.22, 6.64)	0.192		−0.631 (−5.57, 0.801)	6.48 (4.82, 7.77)	5.30 (4.43, 6.66)	5.42 (3.95, 6.99)	0.387		−0.669 (−1.60, 1.24)	0.711 ^v^
Normalized muscular electrical activity vastus lateralis right leg knee flexion–extension with both legs	5.88 (5.05, 7.27)	6.57 (5.84, 7.58)	7.59 (6.80, 9.22)	0.344		1.57 (−0.0880, 3.28)	5.43 (4.45, 6.50)	7.61 (5.01, 10.7)	7.72 (6.25, 10.8)	0.269		2.88 (−0.299, 4.92)	0.495 ^w^
Normalized muscular electrical activity vastus medialis right leg knee flexion–extension with both legs	7.21 (4.77, 9.90)	8.05 (5.45, 10.0)	7.07 (4.13, 8.99)	0.278		0.267 (−3.15, 0.796)	6.36 (3.60, 7.72)	6.19 (4.33, 7.58)	5.55 (3.36, 9.23)	0.990		0.173 (−3.42, 2.43)	0.401 ^x^
Normalized muscular electrical activity vastus lateralis left leg knee flexion–extension with left leg	12.5 (9.29, 17.9)	11.7 (7.27, 16.6)	14.1 (11.1, 16.5)	0.549		1.47 (−1.88, 5.33)	12.8 (9.53, 17.7)	11.3 (7.62, 16.7)	15.5 (11.0, 20.6)	0.305		0.703 (−4.08, 3.98)	0.953 ^y^
Normalized muscular electrical activity vastus medialis left leg knee flexion–extension with left leg	11.8 (7.49, 22.5)	10.4 (6.96, 15.0)	9.46 (7,57, 11.3)	0.042 ^z^		−2.49 (−9.17, 0.703)	12.0 (7.85, 19.0)	10.2 (7.0, 13.2)	9.77 (7.11, 12.5)	0.052		−2.34 (−6.51, 2.19)	0.800 ^aa^
Normalized muscular electrical activity vastus lateralis right leg knee flexion–extension with right leg	10.3 (8.15, 14.8)	13.3 (9.95, 15.9)	13.7 (11.6, 16.0)	0.766		1.93 (−2.44, 4.0)	11.9 (7.85, 15.7)	12.8 (10.2, 18.7)	13.5 (11.4, 22.0)	0.939		−0.991 (−2.67, 8.41)	0.830 ^bb^
Normalized muscular electrical activity vastus medialis right leg knee flexion–extension with right leg	15.9 (10.6, 17.7)	14.3 (12.7, 16.6)	11.7 (8.69, 16.7)	0.420		−2.22 (7.52)	12.1 (7.66, 17.3)	11.8 (8.22, 14.5)	9.53 (6.32, 15.5)	0.646		−1.24 (5.17)	0.672 ^cc^
Symmetry normalized muscular electrical activity vastus lateralis knee flexion–extension with both legs	86.4 (52.8, 94.9)	74.8 (61.8, 83.4)	78.6 (66.4, 85.4)	0.863		−0.185 (32.2)	70.0 (54.8, 85.1)	61.5 (53.2, 76.1)	82.7 (78.3, 91.2)	0.006 ^dd^	V2–V3	12.9 (23.1)	0.202 ^ee^
Symmetry normalized muscular electrical activity vastus medialis knee flexion–extension with both legs	56.7 (46.6, 84.4)	68.4 (50.9, 83.6)	81.2 (70.9, 86.3)	0.070		15.1 (24.2)	73.1 (44.8, 92.0)	67.3 (52.6, 79.7)	73.0 (54.4, 81.6)	0.795		2.51 (33.9)	0.245 ^ff^
Symmetry normalized muscular electrical activity vastus lateralis individual knee flexion–extension left and right leg	75.2 (62.1, 90.4)	71.8 (53.6, 84.5)	81.0 (70.2, 90.4)	0.489		6.59 (31.2)	75.1 (51.9, 86.4)	60.5 (49.5, 83.6)	81.0 (72.7, 93.5)	0.042 ^gg^	V2–V3	13.8 (27.7)	0.500 ^hh^
Symmetry normalized muscular electrical activity vastus medialis individual knee flexion–extension left and right leg	58.6 (43.3, 75.4)	65.5 (61.3, 82.5)	82.3 (51.2, 91.9)	0.035 ^ii^		15.0 (25.1)	69.3 (63.6, 89.5)	59.2 (47.2, 76.4)	73.3 (53.2, 85.7)	0.272		−4.60 (32.6)	0.072 ^jj^
Total pain level	3 (1, 6.5)	2 (0, 5)	2 (0, 4.5)	0.294		0.867 (2.42)	4.5 (1.75, 8.25)	4 (1.75, 6.25)	6 (1, 6.75)	0.492		0.500 (3.54)	0.740 ^kk^
Total stiffness level	2 (0, 4)	1 (0, 2)	0 (0, 1.5)	0.005 ^ll^	V1–V2 V1–V3	1 (0, 2)	2.5 (0, 5.5)	1.5 (0, 4)	1 (0, 4)	0.892		0 (0, 0.5)	0.239 ^mm^
Total difficulty level	4 (2, 15)	1 (1, 10.5)	2 (0.5, 6)	0.003 ^nn^	V1–V2 V1–V3	2 (0.5, 4.5)	18.5 (4.75, 24.3)	11 (0.75, 25)	14.5 (2.75, 26.5)	0.184		2 (−2, 6.5)	0.705 ^oo^
Recovery speed	1.06 (0.858, 1.21)	1.10 (0.927, 1.21)	1.09 (0.979, 1.23)	0.016 ^pp^		0.0466 (−0.00479, 0.147)	0.877 (0.742, 1.00)	0.880 (0.791, 1.08)	0.984 (0.789, 1.04)	0.100		0.0289 (0.00158, 0.0996)	0.682 ^qq^

^a^. Patients with positive difference (% increase). ^b^. CG therapeutic adherence: statistically there is a significant difference since the *p* value is 0.022. The statistical non-parametric test used was ANOVA of repeated Friedman measures (non-normality); for the comparison between couples, the Durbin–Conover method was used, finding that there is a significant difference between initial assessment and final assessment since the *p* value is 0.005. ^c^. IG therapeutic adherence: there is a statistically significant difference since the *p* value is 0.007. The statistical non-parametric test used was ANOVA of repeated Friedman measures (non-normality); for the comparison between couples, the Durbin–Conover method was used, finding that there is a significant difference between initial assessment and final assessment since the *p* value is <0.001. ^d^. Therapeutic adherence: *p* value—chi square. ^e^. CG arrangement: there is strictly no significant difference, since the *p* value is 0.050, the non-parametric statistical test used was ANOVA of repeated Friedman measures (non-normality); however, when comparing between pairs with the Durbin–Conover method, it was found that there is a statistically significant difference between initial assessment and intermediate assessment with a *p* value of 0.030, and between initial assessment and final assessment with a *p* value of 0.030. ^f^. IG arrangement: there is a statistically significant difference since the *p* value is 0.018; the non-parametric statistical test used was ANOVA of repeated Friedman measures (non-normality). When comparing pairs using the Durbin–Conover method, it was found that there is a significant difference between initial assessment and final assessment with a *p* value of 0.004. ^g^. Arrangement: *p*-value—Fisher’s exact test. ^h^. IG responsiveness: there is strictly no significant difference, since the *p* value is 0.061, the non-parametric statistical test used was ANOVA of repeated Friedman measures (non-normality); however, when comparing between pairs with the Durbin–Conover method, it was found that there is a statistically significant difference between initial assessment and final assessment with a *p* value of 0.024. ^i^. Receptivity: *p* value—Fisher’s exact test. ^j^. Normalized knee flexion–extension strength with both legs MVIC CG: there is a statistically significant difference since the *p* value is 0.013; the statistical parametric test used was ANOVA of repeated measures (normality); when making a comparison between pairs using Scheffe’s Post Hoc method, it was found that there is a significant difference between initial assessment and intermediate assessment with a *p* value of 0.013. ^k^. Normalized knee flexion–extension strength with both legs MVIC IG: there is a statistically significant difference since the *p* value is <0.001; the statistical parametric test used was ANOVA of repeated measures (normality); when comparing pairs using Scheffe’s Post Hoc method, it was found that there is a significant difference between initial assessment and final assessment with a *p* value of 0.007, and between intermediate assessment and final assessment with a *p* value of 0.033. ^l^. Normalized knee flexion–extension force with both legs MVIC: student’s *p*-t value. ^m^. Normalized knee flexion–extension force with the left leg MVIC CG: there is a statistically significant difference since the *p* value is 0.009; the statistical parametric test used was ANOVA of repeated measures (normality); when making a comparison between pairs using Scheffe’s Post Hoc method, it was found that there is a significant difference between initial assessment and intermediate assessment with a *p* value of 0.007. ^n^. Normalized knee flexion–extension force with the left leg MVIC IG: there is a statistically significant difference since the *p* value is < 0.001; the statistical parametric test used was ANOVA of repeated measures (normality); when making a comparison between pairs using Scheffe’s Post Hoc method, it was found that there is a significant difference between initial assessment and final assessment with *p* value < 0.001, and between intermediate assessment and final assessment with *p* value 0.004. ^o^. Normalized knee flexion–extension force with the left leg MVIC: student’s *p*-t value. ^p^. Normalized knee flexion–extension force with the right leg MVIC CG: there is a statistically significant difference since the *p* value is 0.025; the statistical parametric test used was ANOVA of repeated measures (normality); when making a comparison between pairs using Scheffe’s Post Hoc method, it was found that there is a significant difference between initial assessment and intermediate assessment with a *p* value of 0.006. ^q^. Normalized knee flexion–extension force with the right leg MVIC IG: there is a statistically significant difference since the *p* value is 0.004; the statistical parametric test used was ANOVA of repeated measures (normality); when making a comparison between pairs using Scheffe’s Post Hoc method, it was found that there is a significant difference between initial assessment and final assessment with a *p* value of 0.015. ^r^. Normalized knee flexion–extension force with the right leg MVIC: student’s *p*-t value. ^s^. Symmetry of force between knee flexion–extension with normalized left and right leg MVIC IG: there is a statistically significant difference since the *p* value is 0.015; the non-parametric statistical test used was ANOVA of repeated Friedman measures (non-normality); when making a comparison between pairs using the Durbin–Conover method, it was found that there is a significant difference between initial assessment and final assessment with a *p* value of 0.003, and between intermediate assessment and the final assessment with *p* value 0.036. ^t^. Symmetry of force between knee flexion–extension with normalized left and right leg MVIC: *p*-value - *t* test. ^u^. Normalized muscular electrical activity vastus lateralis left leg knee flexion–extension with both legs: Mann–Whitney *p*-U value. ^v^. Normalized muscular electrical activity vastus medialis left leg knee flexion–extension with both legs: Mann–Whitney *p*-U value. ^w^. Normalized muscular electrical activity vastus lateralis right leg knee flexion–extension with both legs: Mann–Whitney *p*-U value. ^x^. Normalized muscular electrical activity vastus medialis right leg knee flexion–extension with both legs: Mann–Whitney *p*-U value. ^y^. Normalized muscular electrical activity vastus lateralis left leg knee flexion–extension with left leg: Mann–Whitney *p*-U value. ^z^. Normalized muscular electrical activity vastus medialis left leg knee flexion–extension with left leg CG: there is a statistically significant difference since the *p* value is 0.042; the statistical parametric test used was ANOVA of repeated measures (normality); when making a comparison between pairs using Scheffe’s Post Hoc method, no significant difference was found between the evaluations. ^aa^. Normalized muscular electrical activity vastus medial left leg knee flexion–extension with left leg: Mann–Whitney *p*-U value. ^bb^. Normalized muscular electrical activity of the vastus lateralis right leg knee flexion–extension with the right leg: *p*-U value of Mann–Whitney. ^cc^. Normalized muscular electrical activity vastus medialis right leg knee flexion–extension with right leg: *p*-t value of student. ^dd^. Symmetry normalized muscular electrical activity vastus lateralis knee flexion–extension with both legs IG: there is a statistically significant difference since the *p* value is <0.006; the statistical parametric test used was ANOVA of repeated measures (normality); when making a comparison between pairs using Scheffe’s Post Hoc method, it was found that there is a significant difference between intermediate assessment and final assessment with a *p* value of 0.006. ^ee^. Symmetry normalized electrical muscle activity vastus lateralis knee flexion–extension with both legs: *p*-t value of student. ^ff^. Symmetry normalized muscular electrical activity vastus medialis knee flexion–extension with both legs: *p*-t value of student. ^gg^. Symmetry normalized muscular electrical activity vastus lateralis knee flexion–extension individual left and right leg IG: there is a statistically significant difference since the *p* value is <0.042; the statistical parametric test used was ANOVA of repeated measures (normality); when making a comparison between pairs using Scheffe’s Post Hoc method, it was found that there is a significant difference between intermediate assessment and final assessment with a *p* value of 0.018. ^hh^. Symmetry normalized muscular electrical activity vastus lateralis flexion–extension of individual knee left and right leg: *p*-t value of student. ^ii^. Symmetry normalized electrical muscle activity vastus medialis individual knee flexion–extension left and right leg CG: there is a statistically significant difference since the *p* value is 0.035; the statistical parametric test used was ANOVA of repeated measures (normality); when making a comparison between pairs using Scheffe’s Post Hoc method, no significant difference was found between the evaluations. ^jj^. Symmetry normalized muscular electrical activity vastus medialis individual knee flexion–extension left and right leg: Student’s *p*-t value. ^kk^. Total level of pain: student’s *p*-t value. ^ll^. Total level of rigidity CG: there is a statistically significant difference since the *p* value is 0.005; the non-parametric statistical test used was ANOVA of repeated Friedman measures (non-normality); when making a comparison between pairs using the Durbin–Conover method, it was found that there is a significant difference between initial assessment and intermediate assessment with a *p* value of 0.015, and between initial assessment and the final assessment with *p* value < 0.001. ^mm^. Total stiffness level: Mann–Whitney *p*-U value. ^nn^. Total level of difficulty CG: there is a statistically significant difference since the *p* value is 0.003; the non-parametric statistical test used was ANOVA of repeated Friedman measures (non-normality); when making a comparison between pairs using the Durbin–Conover method, it was found that there is a significant difference between initial assessment and intermediate assessment with a *p* value of 0.015, and between initial assessment and the final assessment with *p* value < 0.001. ^oo^. Total level of difficulty: *p* value – Mann–Whitney U. ^pp^. CG recovery speed: there is a statistically significant difference since the *p* value is 0.016; the statistical parametric test used was ANOVA of repeated measures (normality); when making a comparison between pairs using Scheffe’s Post Hoc method, no significant difference was found between the evaluations. ^qq^. Speed of recovery: Mann–Whitney *p*-U value.

**Table 6 sensors-23-09114-t006:** Summary of results.

Characteristic	GC N = 15 Median (IQR)			*p*-Value	Comparisons between Pairs	Δ GC Mean (Standard Deviation); n (%) ^a^; Median (IQR)	GI N = 16 Median (IQR)			*p*-Value	Comparisons between Pairs	Δ GI Mean (Standard Deviation); n (%) ^a^; Median (IQR)	*p*-value Δ GC vs. Δ GI
	Initial assessment = V1	Intermediate assessment = V2	Final assessment = V3				Initial assessment = V1	Intermediate assessment = V2	Final assessment = V3				
Therapeutic adherence	91.7 (89.2, 100)	93.3 (90, 100)	93.3 (90, 100)	0.022 ^b^	V1–V3	5 (33.3%)	94.2 (91.7, 100)	96,7 (91.7, 100)	98.3, (93.3, 100)	0.007 ^c^	V1–V3	6 (37.5%)	0.809 ^d^
Normalized knee flexion–extension force with both legs MVIC	8.26 (6.67, 11.2)	9.41 (8.18, 11.7)	10.7 (8.46, 11.8)	0.013 ^e^	V1–V2	1.22 (2)	7.83 (5.58, 8.98)	8.98 (6.10, 10.7)	9.40 (6.98, 11.3)	<0.001 ^f^	V1–V3 V2–V3	1.25 (1.34)	0.963 ^g^
Symmetry of force between knee flexion-extension with normalized left and right leg MVIC	88.7 (85.5, 89.9)	89.5 (85.5, 97.4)	90 (86.9, 97.3)	0.420		4.94 (9.69)	84.5 (73.1, 96.4)	90.9 (81.6, 94.9)	91.4 (86.9, 98.8)	0.015 ^h^	V1–V3 V2–V3	8.63 (9.30)	0.287 ^i^
Normalized muscular electrical activity vastus lateralis left leg knee flexion–extension with left leg	12.5 (9.29, 17.9)	11.7 (7.27, 16.6)	14.1 (11.1, 16.5)	0.549		1.47 (−1.88, 5.33)	12.8 (9.53, 17.7)	11.3 (7.62, 16.7)	15.5 (11.0, 20.6)	0.305		0.703 (−4.08, 3.98)	0.953 ^j^
Symmetry normalized muscular electrical activity vastus lateralis knee flexion-extension with both legs	86.4 (52.8, 94.9)	74.8 (61.8, 83.4)	78.6 (66.4, 85.4)	0.863		−0.185 (32.2)	70.0 (54.8, 85.1)	61.5 (53.2, 76.1)	82.7 (78.3, 91.2)	0.006 ^k^	V2–V3	12.9 (23.1)	0.202 ^l^
Total pain level	3 (1, 6.5)	2 (0, 5)	2 (0, 4.5)	0.294		0.867 (2.42)	4.5 (1.75, 8.25)	4 (1.75, 6.25)	6 (1, 6.75)	0.492		0.500 (3.54)	0.740 ^m^
Total stiffness level	2 (0, 4)	1 (0, 2)	0 (0, 1.5)	0.005 ^n^	V1–V2 V1–V3	1 (0, 2)	2.5 (0, 5.5)	1.5 (0, 4)	1 (0, 4)	0.892		0 (0, 0.5)	0.239 ^o^
Total difficulty level	4 (2, 15)	1 (1, 10.5)	2 (0.5, 6)	0.003 ^p^	V1–V2 V1–V3	2 (0.5, 4.5)	18.5 (4.75, 24.3)	11 (0.75, 25)	14.5 (2.75, 26.5)	0.184		2 (−2, 6.5)	0.705 ^q^
Recovery speed	1.06 (0.858, 1.21)	1.10 (0.927, 1.21)	1.09 (0.979, 1.23)	0.016 ^r^		0.0466 (−0.00479, 0.147)	0.877 (0.742, 1.00)	0.880 (0.791, 1.08)	0.984 (0.789, 1.04)	0.100		0.0289 (0.00158, 0.0996)	0.682 ^s^

^a^. Patients with positive difference (% increase). ^b^. CG therapeutic adherence: statistically there is a significant difference since the *p* value is 0.022. The statistical non-parametric test used was ANOVA of repeated Friedman measures (non-normality); for the comparison between couples, the Durbin–Conover method was used, finding that there is a significant difference between initial assessment and final assessment since the *p* value is 0.005. ^c^. IG therapeutic adherence: there is a statistically significant difference since the *p* value is 0.007. The statistical non-parametric test used was ANOVA of repeated Friedman measures (non-normality); for the comparison between couples, the Durbin–Conover method was used, finding that there is a significant difference between initial assessment and final assessment since the *p* value is <0.001. ^d^. Therapeutic adherence: *p* value—chi square. ^e^. Normalized knee flexion–extension strength with both legs MVIC CG: there is a statistically significant difference since the *p* value is 0.013; the statistical parametric test used was ANOVA of repeated measures (normality); when making a comparison between pairs using Scheffe’s Post Hoc method, it was found that there is a significant difference between initial assessment and intermediate assessment with a *p* value of 0.013. ^f^. Normalized knee flexion–extension strength with both legs MVIC IG: there is a statistically significant difference since the *p* value is <0.001; the statistical parametric test used was ANOVA of repeated measures (normality); when comparing pairs using Scheffe’s Post Hoc method, it was found that there is a significant difference between initial assessment and final assessment with a *p* value of 0.007, and between intermediate assessment and final assessment with a *p* value of 0.033. ^g^. Normalized knee flexion–extension force with both legs MVIC: student’s *p*-t value. ^h^. Symmetry of force between knee flexion–extension with normalized left and right leg MVIC IG: there is a statistically significant difference since the *p* value is 0.015; the non-parametric statistical test used was ANOVA of repeated Friedman measures (non-normality); when making a comparison between pairs using the Durbin–Conover method, it was found that there is a significant difference between initial assessment and final assessment with a *p* value of 0.003, and between intermediate assessment and the final assessment with *p* value 0.036. ^i^. Symmetry of force between knee flexion–extension with normalized left and right leg MVIC: *p*-value - *t* test. ^j^. Normalized muscular electrical activity vastus lateralis left leg knee flexion–extension with left leg: Mann–Whitney *p*-U value. ^k^. Symmetry normalized muscular electrical activity vastus lateralis knee flexion–extension with both legs IG: there is a statistically significant difference since the *p* value is <0.006; the statistical parametric test used was ANOVA of repeated measures (normality); when making a comparison between pairs using Scheffe’s Post Hoc method, it was found that there is a significant difference between intermediate assessment and final assessment with a *p* value of 0.006. ^l^. Symmetry normalized electrical muscle activity vastus lateralis knee flexion–extension with both legs: *p*-t value of student. ^m^. Total level of pain: student’s *p*-t value. ^n^. Total level of rigidity CG: there is a statistically significant difference since the *p* value is 0.005; the non-parametric statistical test used was ANOVA of repeated Friedman measures (non-normality); when making a comparison between pairs using the Durbin–Conover method, it was found that there is a significant difference between initial assessment and intermediate assessment with a *p* value of 0.015, and between initial assessment and the final assessment with *p* value < 0.001. ^o^. Total stiffness level: Mann–Whitney *p*-U value. ^p^. Total level of difficulty CG: there is a statistically significant difference since the *p* value is 0.003; the non-parametric statistical test used was ANOVA of repeated Friedman measures (non-normality); when making a comparison between pairs using the Durbin–Conover method, it was found that there is a significant difference between initial assessment and intermediate assessment with a *p* value of 0.015, and between initial assessment and the final assessment with *p* value < 0.001. ^q^. Total level of difficulty: *p* value – Mann–Whitney U. ^r^. CG recovery speed: there is a statistically significant difference since the *p* value is 0.016; the statistical parametric test used was ANOVA of repeated measures (normality); when making a comparison between pairs using Scheffe’s Post Hoc method, no significant difference was found between the evaluations. ^s^. Speed of recovery: Mann–Whitney *p*-U value.

## Data Availability

Results can be found in OneDrive folder: MDPI—Results (https://upbeduco-my.sharepoint.com/personal/vera_perez_upb_edu_co/_layouts/15/onedrive.aspx?id=%2Fpersonal%2Fvera%5Fperez%5Fupb%5Fedu%5Fco%2FDocuments%2Festudiantes%2F202010%5Fjuan%5Fpablo%5Ffonnegra%2FMDPI%20%2D%20Results&ct=1697608482806&or=Teams%2DHL&ga=1, accessed on 15 July 2022).

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
