# Peer review of "Rehabilitation of Patients with Arthrogenic Muscular Inhibition in Pathologies of Knee Using Virtual Reality"

_sensors, 2023, doi:10.3390/s23229114_

Round 1

Reviewer 1 Report

In this manuscript, a rehabilitation system for patients with Arthrogenic Muscle Inhibition (AMI) using virtual reality is presented.

Major Revisions:

Abstract:

The study results, preferably in percentage format, are required for a clearer understanding of whether the intervention improved the patients' condition. Additionally, the main conclusion of the study should be stated. References should not be included in the abstract and it should be compact in a single paragraph.

Introduction:

The second paragraph (lines 47-48) is relatively short; paragraphs should ideally consist of at least four lines. Consider integrating this information into another paragraph. 

The logical flow between the first three paragraphs needs revision. 

The discussion of causes of musculoskeletal disorders in paragraph three (lines 50-53) should be presented in the first paragraph. 

Regarding new techniques like virtual reality (lines 62-65), it is essential to provide context by discussing the use of other emerging technologies. 

The state of the art should be expanded. 

While there might be no studies in Latin America, it would be valuable to explore similar approaches from other parts of the world, as the use of virtual reality in rehabilitation has become widespread in the last five years.

Methodology:

Exclusion criteria: Please clarify the rationale for excluding mental health conditions and define what qualifies as a mental health pathology. 

The formation of the experimental group based on individuals' availability for at least 30 minutes is not maintaining randomness, as it allows patients to self-select into the control group based on their time availability, potentially introducing bias into the results. Randomization entails an assignment that does not consider individual characteristics. 

The means of the groups presented in Table 2 indicate dissimilar conditions at the study's outset, which could influence the results.

To gauge the importance of the variables that constitute the recovery rate, 14 physiotherapists were surveyed. What led to the determination that 14 individuals were sufficient? 

Nothing related to the developed virtual reality application (flowcharts, scenarios, three-dimensional objects, make and model of glasses used) was found. Even in the supplementary videos, only individuals (with very good physical condition) are seen performing exercises in a gym, which is unrelated to virtual reality. I suggest incorporating more evidence, as the current document primarily presents physical rehabilitation. 

Results:

In interpreting Table 5, only relevant results need to be highlighted. Re-explaining the entire table is unnecessary. 

Evidence of the use of virtual reality is required

Discussion:

The discussion mentions that the use of virtual reality could lead to greater adherence in rehabilitation processes, a claim supported by current literature. However, since the specific virtual reality application is not described, this assertion cannot be verified. The discussion should also address practical and knowledge-related implications.

Conclusions:

The conclusions should align better with the presented results (virtual reality) and include limitations and directions for future research.

Minor Revisions:

Materials and Methods:

Equations should be referenced in the text, similar to figures, tables, and annexes, by placing the corresponding number in parentheses. 

I recommend using equation management software for better quality. Some tables have a different font; these should be standardized.

Reviewer 2 Report

Dear authors,

Thank you for sharing your research. Attached you will find a file with some remarks. The results section and the quality of the paper results would improve if a table where the results were summarized was constructed. The reader loses himself with the almost equal sentences for each case.

Round 2

Reviewer 1 Report

Dear authors,

Thank you very much for addressing the suggested changes in your manuscript. Your revisions have substantially improved the quality of the paper. I have just two remaining observations that I believe should be addressed before the article is ready for publication.

I respect your perspective on not including flowcharts regarding the VR application design. However, I would like to emphasize the importance of considering how potential readers will replicate your study without this information. I strongly recommend including these flowcharts unless the editors deem them unnecessary for publication.

Additionally, while the videos provide support for the experiments, it would be pertinent to include images of the virtual environments and participants using the application. This visual representation can enhance the clarity and comprehensibility of your findings.

Regarding the minor changes suggested, I had recommended using an equation editor, but it appears that this was not implemented, and the quality of the equations remains reduced. I strongly recommend using an equation editor to improve the quality of the equations.

Thank you for your attention to these remaining matters, and I look forward to the final revisions for the publication of your article.
